# Associations between Traumatic Experience and Resilience in Adolescent Refugees: A Scoping Review

**Solomon D. Danga** [1,*] **, Babatope O. Adebiyi** [1] **, Erica Koegler** [2] **, Conran Joseph** [3] **and Nicolette V. Roman** [1]

1. Centre for Interdisciplinary Studies of Children, Families and Society, Faculty of Community and Health Sciences, University of the Western Cape, Private Bag x17, Bellville 7535, South Africa
2. School of Social Work, University of Missouri, St. Louis, MO 63121, USA
3. Division of Physiotherapy, Faculty of Medicine and Health Sciences, Stellenbosch University, Tygerberg, Cape Town 7505, South Africa
* Correspondence: 3746783@myuwc.ac.za

**Abstract:** Research on adolescent refugee resilience is crucial for understanding the mechanisms of adaptation to resettlement areas and integration into a new country. However, the current literature does not provide clear evidence on the determinants of resilience factors and the association between traumatic experiences and resilience among adolescent refugees. Four electronic databases were searched to identify relevant articles. Inclusion criteria for articles were (i) potential traumatic experience was the independent variable and resilience was an outcome variable of the study, (ii) association between traumatic experiences and resilience was reported, (iii) participants of the study included adolescent refugees or asylum seekers and (iv) to be peer-reviewed publications based on primary data, written in English and published between 1 January 2010 and 20 January 2022. Eight articles were included in this scoping review. The review found that most of the included studies identified individual, relational/family and contextual/cultural factors as determinants of resilience. However, there were inconsistencies in the association between traumatic experiences and resilience. This review suggests that intervention strategies implemented among adolescent refugees should focus on enhancing individual, family/relational, and cultural/social factors to protect adolescents from possible poor mental health consequences after exposure to trauma.

**Keywords:** potential traumatic experience; resilience; refugees; asylum-seeker; adolescent

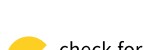

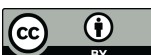

## 1. Introduction

Adolescents and youth refugees face numerous risk factors that impact their wellbeing due to experiences before, during, or after the migration [1–3]. Foster [4] identifies four stages of migration where traumatic experiences may lead to serious psychological distress: (i) premigration traumatic events experienced before migration from the home country; (ii) traumatic experiences during transit to a new country; (iii) continuing traumatic experiences during the process of asylum-seeking and resettlement; and (iv) poor living conditions in the host country due to unemployment, inadequate support and minority persecution. These experiences are often repeated, lengthy, and interpersonal in nature and have negatively impacted mental health conditions [5–9]. Additionally, Nickerson et al. Ref [7] reported that the extent of exposure to traumatic events might vary across several factors, including area/country of origin, characteristics of conflict, and demographic factors such as gender, age, ethnicity and sexual orientation of refugees and asylum-seekers. Traumatic events are one of the risk factors that can affect adolescent refugee health and wellbeing. Traumatic events include war, torture, rape, violent assault, forced labour and serious accidents [4,10]. Children and adolescent refugees in particular migrate with histories of exposure to trauma. Trauma experienced by young refugees may include the violent death of a parent, injury to or torture of a family member, separation from parents, the disappearance of loved ones, enduring political oppression, deprivation of human rights

and education, witnessing murder or massacre, exposure to bombardments, terrorist attack, forcible eviction from home and detention [11]. Trauma exposure related to migration can adversely influence child and adolescent long-term social, emotional, and physical development and well-being [12,13]. However, studies also reported that young refugees are resilient despite experiencing significant adversities such as trauma [14–17].

Researchers in the field of resilience agree that resilience is a complex construct that may be defined differently in the context of individuals, families, organisations, societies, and cultures [18]. From a developmental psychology perspective, Masten [19] defined resilience as individuals having good outcomes despite serious threats to adaptation or development. Masten [20] also describes resilience as the process of positive adaptation and/or recovery from trauma or adversity. Rutter [21] describes resilience as overcoming stress or adverse psychosocial risk experiences in psychiatry. In this scoping review, we operationalised resilience as an individual's competence to maintain and promote good mental health outcomes throughout the course of displacement despite having experienced trauma. We also viewed resilience as not just a personality trait or attribute of an individual [22,23], but rather, it is viewed as a process that refers to exposure to adversity and positive adaptation [24]. Resilience factors for traumatic experiences are commonly identified across three levels of functioning (i) individual factors: related to individuals' psychological and neurobiological conditions such as temperament, learning strengths, self-concept, emotions and social skills, (ii) family factors: related to the family situation such as attachment, communication, parent relations, parenting style and support outside the family, and (iii) social environment factors: related to social conditions, inclusion, access and involvement [25,26]. Youth resilience is determined by the interactions of the individual, family and social factors over time after experiencing trauma [25].

Of the 82.4 million forcibly displaced refugees worldwide (United Nations Higher Commissioner for Refugees [UNHCR] [27], nearly 26.4 million were younger than 18 years. Children and adolescents are the most vulnerable group among refugee populations, of whom many have experienced extreme trauma and negative mental health outcomes [9]. Adolescent refugees are particularly affected by traumatic experiences due to incomplete biological, cognitive and psychological developmental changes and underdeveloped coping skills [11]. Forced migration during adolescence has been found to be riskier than any other period of an individual's life [28]. Studies on the relationship between traumatic experience and resilience among young refugees have been inconclusive. For example, a study conducted among adolescent refugees in South Australia found no significant correlation between the number of traumatic events experienced and resilience scores [29]. However, research on adolescents in the Gaza strip found a statistically significant negative relationship between traumatic events and resilience [30].

Research on the resilience of adolescent refugees is crucial to understanding the mechanisms of adaptation in resettlement areas and for integration into a new country. However, the current literature does not provide clear evidence on the determinants of resilience factors (individual, family, and social environment) for the mental health of adolescent refugees after trauma exposure or the relationship between trauma and resilience. For this reason, a scoping review was conducted to systematically examine the scope and nature of evidence in this research area. Specifically, the following research questions are addressed: (1) What resilience factors determine young refugees' resilience after being exposed to trauma? (2) What is the relationship between traumatic experience and resilience in adolescent refugees?

## 2. Materials and Methods

The PRISMA- Extension for Scoping Reviews (PRISMA-ScR) is a reporting guideline for scoping reviews. This reporting guideline contains 20 essential reporting items and two optional items to be included when conducting a scoping review [31]. Therefore, we adopted the PRISMA Extension for Scoping Reviews (PRISMA-ScR) statements to report the results using 20 essential items in this scoping review.

## *2.1. Eligibility*

Articles from all study designs were included if they met the following inclusion criteria:

(i)     Traumatic experience was the independent variable of the study,
(ii)    Resilience was an outcome variable of the study,
(iii)   Relationship between traumatic experience and resilience was reported,
(iv)    Articles had to be peer-reviewed and published in English based on primary data. Articles were published between 1 January 2010, and 20 January 2022. This period is selected because it is called the world refugee crisis time, and most people are unprecedentedly displaced from their country of origin all over the world.
(v)     Participants of the study included adolescent refugees or asylum seekers. Adolescents aged between 12–18 of all genders were included in this review.

## *2.2. Search Strategy*

This review was based on four electronic databases {Ebsco Host embedded databases (Academic Search Complete, CINAHL Plus with Full Text, SoINDEX, Health Source: Nursing/Academic Edition, Medline and Psyc ARTICLES), PubMed, WoS Core collection, and SCOPUS}. Databases were searched between 15 January 2022 and 5 March 2022 to identify relevant peer-reviewed articles. Included articles reference lists were also searched for additional information.

The search included the following terms alone and in various combinations: "trauma OR traumatic event OR exposure OR experience OR violence OR war trauma OR torture" AND "resilience OR resiliency OR resilient OR mechanism OR strength OR coping OR hardiness OR adaptation" AND "adolescent OR youth OR teen OR teenagers" AND "refugee OR asylum-seeker". The combination of search terms for one database is presented as Supplementary Material S1.

## *2.3. Data Extraction*

Data were extracted using rayyan.qcri software and exported into an Excel spreadsheet. We extracted the following data on articles: primary author and year of publication, source country, study design, participant characteristics, sample size and sampling technique and results based on the inclusion criteria. Two reviewers (SD and AB) searched all the potential articles from the selected databases and independently screened all the potentially relevant titles and abstracts of the articles identified through search strategies. Another two reviewers read the full texts and extracted the data independently (SD and NR). Uncertainty was resolved via a discussion with one of the reviewers (CJ). Finally, another reviewer (EK) verified all the data presented in Tables 1 and 2.

**Table 1.** Description of included Studies Demographic characteristics.

| Author(s) | Country | Study Design | Participant Characteristics and Origin | Sample Size and Sampling Technique |
|---|---|---|---|---|
| Dehnel et al. [32] | Jordan | Cross-sectional | Syrian refugee children. Mean age of 13.4 years, range 10–17. Male (25.7%) and female (74.3%). | N = 339, Non-random sampling |
| Guido et al. [33] | Jordan | Cross-sectional | Syrian children living in Jordanian refugee camps. Mean age 10.5 years, range 7–14. Male (49.8%) and female (50.2%). | N = 311 Random sampling |
| Kim et al. [34] | South Korea | Cross-sectional | North Korean refugee youth. Mean age 18.2 years, range 13–21. Male (51.4%) and female (48.6%). | N = 144, Non-random sampling |

**Table 1.** *Cont.*

| Author(s) | Country | Study Design | Participant Characteristics and Origin | Sample Size and Sampling Technique |
|---|---|---|---|---|
| Mahamid [35] | Palestine (West Bank) | Case study | Palestine adolescent refugees. Mean age 14.8 years, range 14–16. Male (43.3%) and female (*n* = 56.7%). | N = 30, Snowball sampling |
| O'connor & Seager [36] | Bangladesh | Cross-sectional | Rohingya adolescents living in camps. Mean age 16 years, range 15–18. Male (49%) and female (51%). | N = 361 Random sampling |
| Dangmann et al. [37] | Norway | Cross-sectional | Syrian Refugee youth. The mean age was 18 years, range 12–24. Male (62.5%) and female (37.5%). | N = 161 Strategic sampling |
| Sleijpen et al. [38] | Netherlands | Case study | Treatment-seeking refugee youth originating from the Middle East, Africa, Eastern Europe, and Asia. Mean age 16.7 years, range 13–21. Male (50%) and female (50%). | N = 16 Non-random sampling |
| Uysal et al. [39] | Turkey | Cross-sectional | Syrian adolescent refugees. Mean age 15.5 years, range 12–18. Male (44%) and female (56%). | N = 430 Random sampling |

**Table 2.** Studies results on resilience factors measured and the relationship between traumatic experience and resilience.

| Author(s) | Resilience Factors Measured | Relationship between Traumatic Experience and Resilience |
|---|---|---|
| Dangmann et al. [37] | Individual, relational and contextual dimensions. | Potential traumatic events were negatively correlated with resilience (r = −0.20, *p* < 0.05). |
| Dehnel et al. [32] | Individual (personal and social skills), relational (such as child's social support) and contextual factors (spirituality and environmental influences). | Traumatic life events were not significantly correlated with resilience. |
| Guido et al. [33] | Personal skills, social resources and contextual factors. | Higher levels of trauma exposure were correlated with lower resilience (r = −0.213, *p* < 0.001). |
| Kim et al. [34] | Individual resiliency (e.g., hardiness, intimacy and clear sense of goals). | Trauma exposure was not significantly associated with ego resiliency (R = 0.02) |
| Mahamid [35] | Individual resiliency (e.g., positive self-efficacy, effective coping, psychological hardness and responsibility). | Children in the sample group expressed high levels of resiliency in dealing with traumatic and painful experiences. |
| O'connor & Seager [36] | Individual, relational, communal and cultural. | Exposure to traumatic events was positively and significantly associated with resilience |
| Sleijpen et al. [38] | Individual and social contexts factors such as (1) acting autonomously, (2) performing at school, (3) perceiving support from peers and parents and (4) participating in the new society | Resilience helped young refugees strengthen their sense of power and control, give them some distraction, and support or sustain their spirit within the family unit and the new society. Almost all of them (*n* = 15) found that they had psychologically matured: they had become stronger and more independent through their hardships. |
| Uysal et al. [39] | Individual, relational, communal and cultural. | Resilience was negatively correlated with occurrence of traumatic events (r = −0.34, *p* < 0.001) and negative appraisal of trauma (r = −0.26, *p* < 0.01). |

*2.4. Data Synthesis*

Data are presented in a tabular form. All the included studies were grouped based on the types of information they analysed and summarised based on the descriptive characteristics of the studies (primary author and year of publication, source country, study design, participant characteristics, sample size and sampling technique) and results based on the inclusion criteria.

## 3. Results

A total of 749 articles were identified as potentially relevant for this study. After the removal of duplications, 421 studies were screened for title and abstract. Based on title and abstract screening, 17 articles were considered relevant for further full-text reading and extraction. Then, based on the inclusion and exclusion criteria, eight articles were included in this scoping review. Figure 1 presents a flow diagram of the search and screening process.

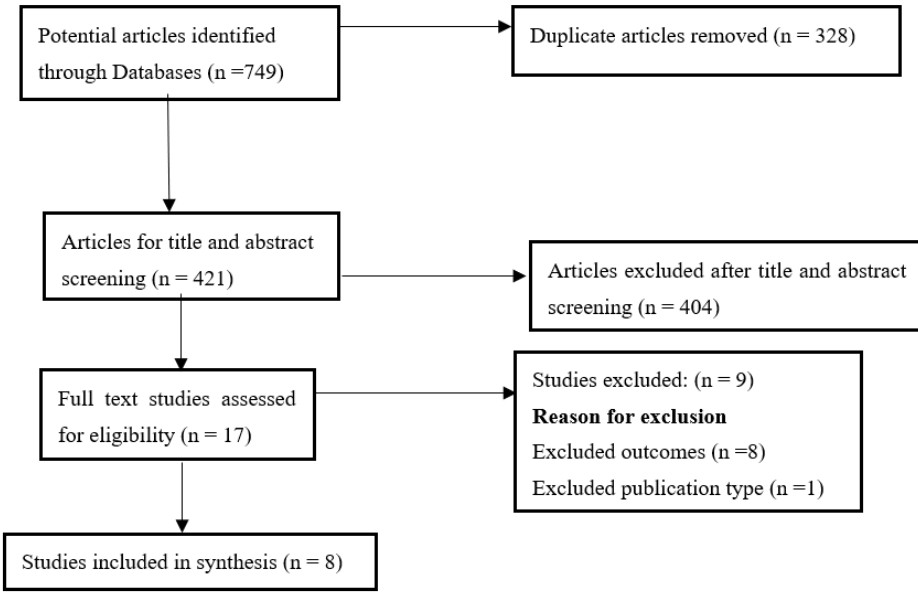

**Figure 1.** Flow diagram of the scoping review study selection process.

Table 1 presents the following data from the included articles: author (s), the country where the study took place, study design, participant characteristics, country of origin, and sample size and sampling technique. Five studies were conducted in Asian countries [32–36], and three studies were from Europe [37–39]. Six studies employed a cross-sectional research design [32–34,36,37,39] and two studies [35,38] utilised a case study. Half of the included studies (*n* = 4) employed forms of random sampling, and the remaining studies (*n* = 4) used non-random sampling techniques.

The majority of the studies (*n* = 7) included participants from Asian countries (i.e., Syria, North Korea, Myanmar, Palestine) and one study included participants from the Middle East, Africa, Eastern Europe, and Asia. In terms of the participants of the study, a total of 1792 participants were included across the included studies, of which more than half (*n* = 988) were female participants. The smallest study included 16 participants, and the largest study included 430 participants.

Table 2 provides information for each included article on determinants of resilience factors and results for the relationship between traumatic experience and resilience among adolescent refugees.

*3.1. Determinants of Resilience Factors in Adolescent Refugees*

Studies included in this study employed various resilience measures and ways to assess resilience factors. Most of the included studies (*n* = 5) reported that the interaction of individual, relational/family, and contextual/cultural resilience factors determined the

adolescent refugee's resilience [32,33,36,37,39]. Two included studies reported individual resiliency factors such as hardiness, intimacy, a clear sense of goals, positive self-efficacy, effective coping and responsibility [34,35]. One study reported individual and social context determinants of resilience factors such as acting autonomously, performing at school, perceiving support and parents, and participating in the new society [38].

### 3.2. Relationship between Traumatic Experience and Resilience

Studies included in this review measured resilience factors differently. This difference in the use of different resilience factors may be due to the theoretical positions of the researchers in measuring resilience. The included studies' findings on the relationship between traumatic experiences and resilience among adolescent refugees were inconclusive. This inconclusive result may be attributed to the difference in the research methods (random sampling and large sample size) and measurements used to assess the resilience factors. Among the included studies, three articles found a negative correlation between traumatic experience and resilience of the participants—traumatic experience was associated with lower resilience of the participants [33,37,39]. Three articles showed positive association between traumatic experiences and resilience—trauma was associated with higher resilience [35,36,38], and two studies reported no significant association between traumatic experiences and resilience among adolescent refugees [32,34]. Studies conducted on Syrian youth refugees in Norway, Jordan and Turkey found that traumatic experiences were negatively associated with resilience [33,37,39] (Dangmann et al., 2021; Guido et al., 2021; Uysal et al., 2022). However, studies conducted in Palestine, Bangladesh and Netherland on young refugees found a positive association between traumatic experiences and resilience factors [35,36,38]. Dehnel et al. [32] and Kim et al. [34] reported that traumatic exposure was not significantly associated with resilience among adolescent refugees.

## 4. Discussion

As Southwick et al. [18] (2014) stated, resilience is a complex construct that may be defined differently in the context of individuals, families, organisations, societies, and cultures. Studies included in this review defined and measured resilience differently. Some included studies defined and measured resilience as an individual trait, while most studies considered resilience an ongoing process resulting from the interactions between individuals and social and cultural factors.

This scoping review included eight studies identifying resilience factors and the association between traumatic experience and resilience in adolescent refugees published between 1 January 2010, and 20 January 2022. This limited number of studies implied a dearth of research examining the association between traumatic experiences and resilience among adolescent refugees. Most of the included studies identified individual, relational/family and contextual/cultural factors as determinants of resilience. This finding agreed with previous studies, which identified three levels of factors that determine resilience among the youth population: (i) individual factors: related to individual psychological and neurobiological conditions such as temperament, learning strengths, self-concept, emotions and social skills, (ii) family factors: related to the family situation such as attachment, communication, parent relations, parenting style and support outside the family, and (iii) social environment factors: related to social conditions, inclusion, access and involvement [25,26]. Barankin and Khanlou [25] also reported that young people's resilience is determined by the interplay between individual, family and social factors over time after experiencing trauma. However, two studies reported ego resiliency as the determinant of resilience in adolescent refugees [34,35]. Again, this finding is supported by previous studies which indicated resiliency as an individual's personality trait or attribute [22,23].

The extent and nature of exposure to traumatic events may vary across several factors, including area/country of origin, characteristics of conflict, and demographic factors such as gender, age, ethnicity and sexual orientation of refugees and asylum-seekers [7]. Studies included in this review defined and assessed traumatic experiences among adoles-

cent refugees differently. This scoping review showed inconsistencies in the association between traumatic experience and resilience across the included studies. For instance, three studies found a negative correlation between traumatic experiences and participants' resilience [33,37,39]. This finding supports previous research on adolescent refugees in the Gaza strip, which found a statistically significant negative relationship between traumatic events and resilience [30]. However, two studies found no significant association. This result also agreed with Ziaian et al. [29], who found no significant correlation between the number of traumatic events experienced and resilience scores among adolescent refugees in South Australia. Three other studies indicate positive associations [35,36,38]. These three studies' findings implied that as the traumatic experience increased, adolescent refugees' resilience increased and vice versa. These inconsistencies across the included studies may be attributed to variations in adolescent refugees' individual and social contexts before and after forced displacement, and the extent, nature and measurement of traumatic exposure could be contributing factors. Moreover, the small sample size and non-random selection of participants could be factors for the inconsistencies of the results.

Adolescence, the transition period between childhood and adulthood, involves biological, cognitive, and socioemotional changes [40]. Adolescent refugees are particularly affected by traumatic experiences due to incomplete biological, cognitive and psychological developmental changes and underdeveloped coping skills [11]. Forced migration during adolescence has been found to be riskier than any other period of an individual's life [28]. In 2018, children under the age of eighteen constituted roughly half the global refugee population [UNHCR] [41]. Children and adolescent refugees particularly migrate with histories of exposure to trauma. Trauma experienced by young refugees may include the violent death of a parent, injury to or torture of a family member, separation from parents, the disappearance of loved ones, enduring political oppression, deprivation of human rights and education, witnessing murder or massacre, exposure to bombardments, terrorist attacks, forcible eviction from home, and detention [11]. Previous studies have also shown increased rates of mental health concerns among refugees forcibly displaced during adolescence, indicating these young refugees may be a particularly at-risk subgroup within the broader global refugee population [2,42,43]. Despite traumatic events and experiences being identified as risk factors for adolescent refugees' mental health problems, studies also indicated protective and promotive factors such as coping, resilience, self-esteem, self-social support and social cohesion that buffer youth vulnerability to mental health problems [14,44–47]. Resilience, as one of these protective and promotive factors, buffers the effects of potential traumatic experiences among adolescent refugees. Adolescent refugee resilience is determined by the interactions of the individual, family and social factors over time after experiencing trauma [25].

The result of this review also revealed that more than half of the participants in the included studies were female. This shows that female adolescent refugees are more vulnerable to traumatic experiences than male adolescent refugees. This finding supported Ward and Vann [48] study, which stated that women and girls are vulnerable to suffering from sexual violence, including forced sex/rape, sexual abuse by an intimate partner, child sexual abuse, coerced sex and sex trafficking in settings of humanitarian conflicts. Another study also revealed that at refugee settings, the Congolese women and girls are often separated from other family members due to the disruption of social structures by armed conflict, making them more vulnerable to sexual and gender-based violence [49].

## 5. Strengths and Limitations

The strengths of this scoping review include articles searched from four electronic databases, which allowed the reviewers to get adequate articles for the study. Furthermore, most studies reported robust methods (such as large sample size and reliable measurements). However, the findings of this scoping review are subjected to the following limitations. First, the review excluded grey and non-English literature, resulting in the absence of pertinent research conducted in other languages. Second, it did not assess the

included studies' risk of bias, which may affect the transparency of evidence synthesis results from each included article. The lack of longitudinal research on adolescent refugees that met this study's inclusion criteria is one of the limitations of this review. Finally, all the included studies employed cross-sectional and case study designs, which does not allow establishing a cause-effect relationship between trauma and resilience. Therefore, results should be interpreted with caution.

## 6. Recommendations

This review indicates that young refugees' resilience is determined by the interplay of individual, family/relational and cultural/social factors. Thus, intervention strategies implemented for adolescent refugees should focus on enhancing individual, family/relational, and cultural/social factors to protect adolescents from possible poor mental health consequences after exposure to traumatic experiences. However, there were inconsistencies in the association between traumatic experience and resilience in adolescent refugees exposed to trauma as risk factors and resilience as protective factors. Therefore, future research should provide a clear picture of the existing relationship between traumatic experience and resilience to know the scope and nature of these variables' relationship. Additionally, future research should focus on the type and severity of potential traumatic experiences and other risk factors such as social isolation, loneliness, and parental separation of adolescent refugees. Furthermore, longitudinal studies are recommended to investigate the cause-effect relationships between trauma and resilience among adolescent refugees.

International organisations like UNHCR should develop resilience-based intervention guidelines for children and adolescents exposed to traumatic experiences and mental health problems. Mental health professionals working in refugee camps should also focus on enhancing the resilience ability of adolescent refugees. At the policy level, understanding the determinants of resilience factors and the relationship between traumatic experiences and resilience will assist policymakers in integrating resilience-based mental health services in refugee camps.

## 7. Conclusions

This scoping review identified the determinants of resilience factors and examined the relationship between trauma and resilience among adolescent refugees. The review concluded that (1) the interaction of individual, family/relational and cultural/social factors determined adolescent refugee resilience after being exposed to traumatic events. (2) there is no clear evidence in the literature on the association between traumatic experiences and the resilience of adolescent refugees.

**Supplementary Materials:** The following supporting information can be downloaded at: https://www.mdpi.com/article/10.3390/youth2040048/s1, Scoping review search terms for the Ebscohost database attached as Supplementary Material S1.

**Author Contributions:** S.D.D. contributed to the conception and drafted the manuscript, searched and screened the studies and extracted and analysed the data. B.O.A. searched and screened the titles and abstracts of potential articles. E.K. participated in verifying the extracted data and critically reviewed the manuscript. C.J. contributed to resolving uncertainty and critically reviewed the manuscript. N.V.R. participated in the data extraction and critically reviewed the manuscript. All authors have read and agreed to the published version of the manuscript.

**Funding:** This scoping review received no external funding.

**Institutional Review Board Statement:** Not applicable.

**Informed Consent Statement:** Not applicable.

**Data Availability Statement:** All data generated and analysed are included in the manuscript.

**Conflicts of Interest:** The authors declare that they have no conflict of interest.

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
