# Peer review of "Associations between Traumatic Experience and Resilience in Adolescent Refugees: A Scoping Review"

_2673-995X, doi:10.3390/youth2040048_

Round 1

Reviewer 1 Report

It seems to me a good and interesting work. I think it is an important topic and a literature review is always very useful to scholars and readers in general.

I think the work is well-done. I just have some comments about the discussion part. The identified articles are not a lot (just 8, maybe some words commenting this can be interesting - resilience turns up everywhere).  I would suggest the authors to expand the results section (maybe also describing the samples, gender lens etc)  and elaborate a bit more the discussion, which is a little too short and fast. 

Reviewer 2 Report

Overall the paper is well written and well organized. It would be helpful for the the authors to expand on the complexity of the refugee migration experience and understanding of resilience as an ongoing process rather than a trait. By not examining these nuances, we may risk overgeneralization and oversimplification. Furthermore, it is difficult to make any conclusions based on the current findings given the vast differences in how resilience was defined and measured across the sampled studies. It is also unclear whether there was consistency across the studies on what they defined as traumatic experiences which may have also impacted the findings. Cultural differences, context of resettlement, gender, and age could also played an important role in the findings which all should be expanded upon in the discussion section. 
